# Evaluation of the Performance Characteristics of a New POC Multiplex PCR Assay for the Diagnosis of Viral and Bacterial Neuromeningeal Infections

**DOI:** 10.3390/diagnostics13061110

**Published:** 2023-03-15

**Authors:** Hervé Le Bars, Neil Madany, Claudie Lamoureux, Clémence Beauruelle, Sophie Vallet, Christopher Payan, Léa Pilorgé

**Affiliations:** 1Unity of Bacteriology, Department of Bacteriology-Virology-Parasitology-Mycology-Hygiene, Pole of Biology-Pathology, University Hospital of Brest, F-29200 Brest, France; 2Unity of Virology, Department of Bacteriology-Virology-Parasitology-Mycology-Hygiene, Pole of Biology-Pathology, University Hospital of Brest, F-29200 Brest, France; 3Univ Brest, Inserm, EFS, UMR 1078, GGB CEDEX, F-29200 Brest, France

**Keywords:** point of care, syndromic, PCR, cerebrospinal fluid, meningitis, encephalitis, bacteria, virus

## Abstract

Point-of-care syndromic PCR (POC SPCR) assays are useful tools for the rapid detection of the most common causative agents of community-acquired infections responsible for meningitis and encephalitis infections. We evaluated the performance characteristics of the new QIAstat-Dx^®^ Meningitis/Encephalitis panel (QS) compared to the laboratory reference methods and the POC SPCR Biofire^®^ FilmArray^®^ Meningitis Encephalitis Panel (FA). Viral (Enterovirus, Parechovirus, HSV-1, HSV-2, HHV-6, VZV) and bacterial (*E. coli* K1, *H. influenzae*, *L. monocytogenes*, encapsulated *N. meningitidis*, *M. pneumoniae*, *S. agalactiae*, *S. pneumoniae*, *S. pyogenes*) pathogens were suspended at low concentrations and tested with the POC SPCR systems. The reproducibility, analytical specificity, carryover contamination, interferences and clinical samples were evaluated. All samples tested positive with both QS and FA except for those containing the lowest concentrations of Enterovirus-D68-B3, Echovirus-30 and *S. agalactiae* which were only detected by FA. In terms of analytical specificity, we observed 3 false positive results out of 48 QS tests versus 1 out of 37 FA tests. For the other studied criteria, both QS and FA performed as expected. Our results suggest that the performance characteristics of QS are close to those of FA. A prospective multicenter study would be useful to complete the performances evaluation of QS.

## 1. Introduction

Point-of-care syndromic PCR (POC SPCR) assays are useful tools for the detection of the most common causative agents of community acquired infectious meningitis/encephalitis in cerebrospinal fluid (CSF) within less than 2 h. Such a rapid diagnosis is of clinical importance to improve the medical care of the patients suffering from these life-threatening infections [1,2]. Indeed, a rapid initiation of appropriate treatment based on the causative pathogen (e.g., amoxycillin and gentamicin in cases of *Listeria*) is necessary to improve patient outcomes [3]. POC SPCR may be complementary of direct microbiological examination following Gram staining which is less sensitive in cases with low bacterial inoculum and/or antibiotic intake prior to lumbar puncture [3]. Furthermore, in those cases of prior antibiotic intake to lumbar puncture, POC SPCR may be the only way to identify the causative agent since the sensitivity of culture greatly decreases in these cases [3]. POC SPCR may also enable to give a rapid diagnosis of viral meningitis (Enterovirus, Parechovirus) therefore reducing inappropriate antibiotic use [4]. In addition, POC SPCR assays target a large number of pathogens in 200 μL of CSF, thus enabling to spare some of this precious biological matrix. Finally, diagnosis algorithms may be useful to determine when POC SPCR should be used in the laboratory in order to optimize their clinical relevance [5].

In 2022, the new POC SPCR QIAstat-Dx^®^ Meningitis/Encephalitis panel (QS) assay, run-on the QIAstat-Dx analyzer system, allows detection of 15 bacteria, virus and fungal that cause meningitis/encephalitis became available.

In this investigation, we evaluated the performance characteristics of this panel compared to the laboratory reference methods and the previously commercialized POC SPCR BioFire^®^ FilmArray^®^ Meningitis Encephalitis Panel (FA), for the detection of viral and bacterial nucleic acids in CSF. Fungal pathogens were not evaluated in this study.

## 2. Materials and Methods

### 2.1. POC Syndromic PCR Meningitis/Encephalitis Panel

The QS panel (Qiagen, Germany) has the capacity to detect 15 pathogens including 6 viral targets (Herpes simplex HSV-1, HSV-2, Varicella-zoster VZV, enterovirus (EV), parechovirus (PeV), herpesvirus human 6 (HHV-6)), 8 bacterial targets (*Escherichia coli K1* (*Ec*), *Haemophilus influenzae* (*Hi*), *Listeria monocytogenes* (*Lm*), encapsulated *Neisseria meningitidis* (*Nm*), *Streptococcus agalactiae* (GBS), *Streptococcus pneumoniae* (*Sp*), *Streptococcus pyogenes* (GAS) and *Mycoplasma pneumoniae* (*Mp*)), 2 fungal pathogens (*Cryptococcus gattii*/*Cryptococcus neoformans* both detected but not differentiated) and an internal control (IC). QS provides semi-quantitative results by providing access to amplification curves and cycle threshold (Ct) values for all pathogens and the internal control. The FA panel (bioMérieux, Marcy l’Etoile, France) is an FDA-cleared test since 2015. FA contains 14 targets and are the same as those found on QS except for *Streptococcus pyogenes* and *Mycoplasma pneumoniae* which are not included. On the other hand, cytomegalovirus is an additional target present in FA and not found in QS. Reported results with FA are only qualitative. It takes around 1 h to obtain test results for both POC SPCR assays.

### 2.2. CSF Pool Preparation

We used a pool of CSF samples from 100 patients. Each of these CSF samples had the following features: less than 10 leukocytes and 10 red blood cells per µL and no bacterial growth. The pool was also tested to confirm the absence of HSV-1, HSV-2, VZV, HHV-6 and *Mp* DNA and EV and PeV RNA with specific real-time PCR assays (R-GENE^®^, bioMérieux, France). This negative CSF pool was then spiked with virus or bacteria to evaluate the different parameters of the study. The pool dilution in the matrix containing the virus (Universal transport medium UTM) or bacteria (Phosphate buffered solution PBS) was 10% maximum. Spiked CSF aliquots were then kept at −80 °C until analysis.

### 2.3. Virus

We used 4 strains of EV: Enterovirus-A71-C1 (EV-A71-C1), Enterovirus-D68-B3 (Ev-D68-B3), Echovirus-30 (E-30), Echovirus-6 (E-6), and 2 strains of PeV: parechovirus 1 (PeV-1) and 3 (PeV-3). EV-A71-C1, E-30 and PeV-1 were kindly provided by the National Reference Center for enteroviruses (National Reference Center for enteroviruses, Lyon, France). The other EV and PeV strains correspond to external quality controls provided by the international external quality assessment organization Quality Control for Molecular Diagnostics (QCMD). We used clinical strains of HSV-1, HSV-2 and VZV isolated in our laboratory and the World Health Organization international standard for HHV-6B (NIBSC code 15/266).

### 2.4. Bacteria

The bacterial strains used in this study and their origin are described in Table 1. They were either standard reference strains from ATCC and DSMZ or clinical strain isolated in the teaching hospital of Brest (France) and further confirmed by the French national reference centers. *Mycoplasma pneumoniae* M129 was obtained from the laboratory of bacteriology of the university hospital of Bordeaux (France).

### 2.5. Assessed Parameters

#### 2.5.1. Detection of Low Viral and *Mycoplasma pneumoniae* Loads

Our reference method was a specific real-time PCR assay (R-GENE^®^, bioMérieux) for HSV-1, HSV-2, VZV, HHV-6, EV, PeV and *Mp*. Ten-fold serial dilutions were prepared in CSF pool for each virus and *Mp* and aliquots of each dilution were stored at −80 °C. Then, one aliquot of each dilution was thawed at room temperature, nucleic acids were extracted with the eMAG^®^ (bioMérieux, France) and the eluate was tested 5 times with R-GENE^®^ PCR assays. The lowest concentration of viral or *Mp* nucleic acids amplified with R-GENE^®^ PCR assay in 5 cases out of the 5 replicates was chosen for testing once with QS and FA, using a new thawed aliquot. If the QS and FA result was positive, no further testing was performed. If QS or FA result was negative, the previous diluted sample, i.e., 10 times more concentrated, was tested once. In case of a negative result at this dilution, no further testing was performed. In case of positivity, 5 new replicates were tested at the lowest concentration of viral nucleic acids amplified in 100% of cases with the R-GENE^®^ PCR assay. A schematic diagram of this strategy is shown in Figure 1.

#### 2.5.2. Detection of Low Bacterial Concentrations

We evaluated the detection of low bacterial concentrations with QS and FA by testing samples prepared by spiking the CSF pool with CFU/mL quantified suspensions of bacteria to reach the limit of detection (LoD) announced by the manufacturer of the already commercialized FA (1000 CFU/mL for *Escherichia coli*, *Haemophilus influenzae*, *Listeria monocytogenes*, *Streptococcus agalactiae* and 100 CFU/mL for encapsulated *Neisseria meningitidis* and *Streptococcus pneumoniae*). For *Streptococcus pyogenes,* which is only included in the QS panel, a 1000 CFU/mL suspension was analyzed.

#### 2.5.3. False Positive Results

We observed that no pathogen was detected with QS and FA in addition to the spiked pathogen in the CSF pool. We also tested high concentrated (106 CFU/mL) suspensions of three closely related species of *Streptococcus pneumoniae*: *Streptococcus mitis*, *Streptococcus oralis* and *Streptococcus pseudopneumoniae* (Table 1).

#### 2.5.4. Reproducibility and Carryover Contamination

We spiked the CSF pool with HSV-1, PeV-1, *Nm* and *Sp* (Ct values between 28 and 32). Reproducibility was evaluated by performing 3 QS and 3 FA assays with this spiked CSF pool. Carryover contamination was studied by testing 2 cartridges of non-spiked CSF pool between them.

#### 2.5.5. Interferences

We spiked the CSF pool with HSV-1 alone, or both HSV-1 and a potentially interfering agent: 10% of whole blood, 5 g/L of bovine serum albumin (Ambion^®^) or 1500 leukocytes/µL. Whole blood was tested with a specific real-time PCR assay (R-GENE^®^) to confirm the absence of HSV-1, HSV-2, VZV and HHV-6 DNA. The leukocytes were isolated from a urine sample. For the QS assay, we considered interference if the HSV-1 Ct value in the CSF pool with an interfering agent was more than 3 compared to the HSV-1 Ct value in the CSF pool without an interfering agent. For the FA assay, which does not provide Ct values, we considered interference if HSV-1 result in the CSF pool with interfering agent was negative.

#### 2.5.6. Clinical CSF Samples

We tested QS and FA retrospectively with 3 clinical CSF samples known to be positive for HSV-2 (Ct value at 37.2 with BioGX^®^ Viral Meningitis HSV/VZV assay), VZV (Ct value at 33.7 with R-GENE^®^ PCR assay) and PeV (Ct value at 34.3 with R-GENE^®^ PCR assay) and 2 clinical CSF samples which had previously grown *Nm* or *Sp*. For HSV and VZV, these fluids had been stored at −80 °C. For bacteria, CSF samples had been stored at 4 °C and tested less than 48 h after collection.

## 3. Results

### 3.1. Detection of Low Viral and Mycoplasma pneumoniae Loads

R-GENE^®^ PCR assays, QS and FA results and Ct values when available are described in Table 2. The dilution chosen to be tested with QS and FA in the first place is the 1/10 dilution because it was the lowest one that generates 5 out of 5 positive results with R-GENE^®^ PCR assays.

Of the 10 viral targets tested once at 1/10 dilution with QS, all were positive except 3 EV strains. Of these 3 strains, E-30 was positive without dilution and positive in 3 of the 5 new replicates at 1/10 dilution. EV-A71-C1 was only positive without dilution and EV-D68-B3 was not positive at either 1/10 or without dilution. FA detected all viral targets at 1/10 dilution except for EV-D68-B3 which was positive without dilution and positive in 5 of 5 new replicates at 1/10 dilution.

### 3.2. Detection of Low Bacterial Concentrations

For each bacterial target, suspensions at the FA announced LoD were tested. They were all detected by both QS and FA except for GBS which was only detected by FA. With QS, 3000 CFU/mL was the lowest concentration of GBS which tested positive (Table 3).

### 3.3. Analytical Specificity

We performed 48 QS assays and 37 FA assays. False positive results for QS and FA are described in Table 4.

We found four false positive results. In the CSF pool spiked with 200 CFU/mL of *Sp*, HSV-1 was unexpectedly detected by both QS (Ct 35.6) and FA. Furthermore, QS unexpectedly detected *Hi* (Ct 37.5) in the CSF pool spiked with *Lm* and HHV-6 (Ct 37.3) in the HSV-1 spiked CSF pool with 10% whole blood used in the interference’s tests.

No cross reactivity was observed on FA and QS between *Streptococcus mitis, Streptococcus oralis*, *Streptococcus pseudopneumoniae* and the detection of *Streptococcus pneumoniae*.

### 3.4. Reproducibility and Carryover Contamination

The four pathogen targets were 100% positive with both POC SPCR assays for the three cartridges. QS Ct values were very close to each other (below 1 Ct). The two cartridges tested with the negative CSF pool did not detect any viral or bacterial targets.

### 3.5. Interferences

HSV-1 DNA was detected by both QS and FA POC SPCR assays in all four cases and the addition of potentially interfering agents did not significantly shift the QS Ct values (below 3 Ct, from 0.3 to 1 Ct).

### 3.6. Clinical CSF Samples

The five viral and bacterial targets (HSV-2, VZV, PeV, *Nm*, *Sp*) detected in the clinical CSF samples were all found positive with QS and FA (Table 5).

### 3.7. Internal Control Reproducibility

The maximum differences of the internal control Ct values in 2 series of 5 cartridges loaded with the same sample were 3 (E-30 at 1/10 dilution) and 3.8 (EV-A71-C1 at 1/10 dilution).

## 4. Discussion

This study was designed to provide the first analytical assessment of the POC SPCR assay QIAstat Dx^®^ (QS) which has recently been developed by Qiagen for the detection of pathogens that potentially cause central nervous system (CNS) infections.

For HSV-1, HSV-2, VZV, HHV-6, PeV-1, PeV-3, E-6 and *Mycoplasma pneumoniae*, both QS and FA correctly detected the lowest loads included in the study. Prior studies noticed low HSV-1, HSV-2 and VZV DNA concentrations in CSF in cases of encephalitis and meningitis [6,7,8,9,10]. A cut-off of approximately 200 copies/mL is considered to be necessary for accurate diagnostics [11]. In our study, DNA quantification of HSV-1, HSV-2, and VZV in 1/10 dilutions CSF was 1280 copies/mL, 277 copies/mL and <300 copies/mL with R-GENE^®^ PCR assays, respectively. These concentrations are close to the 200 copies/mL cut-off. As a comparison, the 95% LoD of the R-GENE^®^ PCR assays have been determined at 250 copies/mL, 100 copies/mL and 300 copies/mL for HSV-1, HSV-2 and VZV, respectively (bioMérieux’s manufacturing data). Those for FA were recently recalculated by bioMérieux’s research and development department and evaluated at 500 copies/mL for HSV-1 and HSV-2, and 1000 copies/mL for VZV (bioMérieux’s manufacturing data). A recent report with the analysis of 1334 pediatric and 336 adult CSF samples tested with FA describes a sensitivity of 75% for HSV-1 compared to a virus-specific PCR [12]. Of note, QS 95% LoD were expressed in TCID50/mL (supplier’s technical data sheets), so they were not appropriate for molecular biology. Use of an absolute quantification tool to calibrate the quantification standards, such as digital PCR [13] and comparison of replicates results on limit dilutions would allow for a reliable determination of the LoD and relative sensitivity of the two POC SPCR assays. For EV detection PCR assays, the challenge is to include as many genotypes as possible [14]. Indeed, the sensitivity of EV PCR assays depends on the genotypes [15]. Moreover, CSF viral loads vary according to the genotype [16]. In our study, QS did not detect the lowest loads of E-30 and EV-D68-B3 in contrast to the R-GENE^®^ PCR assay and FA. As EV-D68 genotype is mostly detected in peripheral samples and not in CSF samples in case of neuromeningeal symptoms [17], this may not constitute an important pitfall. However, additional testing including a larger number of genotypes would be necessary for a more accurate assessment of EV inclusivity. The study of Schnuriger et al. describes a FA sensitivity of 89% for enterovirus detection compared to a virus-specific PCR [12].

Concerning the Detection of low bacterial concentrations, all the analyzed samples were correctly detected by both QS and FA except GBS. The sample containing 1000 CFU/mL of GBS tested positive with FA only. These results are consistent with the LoD announced by both manufacturers. Previous studies have found false negative results to be very uncommon with FA [18] meaning that in the vast majority of bacterial meningitis, the bacterial loads are higher than FA’s LoD. The LoD announced for both QS and FA are lower for *Sp* and *Nm* than for other targets, which is clinically relevant since *Sp* and *Nm* are the two main species isolated as causative agents of community-acquired bacterial meningitis in France, representing more than 70% of the cases [3]. For GBS, our results suggest a slightly better detection of low concentrations with FA since it was solely able to detect 1000 CFU/mL. However, a 3000 CFU/mL suspension of GBS tested positive with QS which does not seem very far from FA and such a tiny difference may not be clinically relevant for the detection of most GBS meningitis cases since most CSF samples contains more than 1000 CFU/mL in clinical cases [19].

Concerning cross reactivity between close species, *Streptococcus mitis*, *Streptococcus oralis* and *Streptococcus pseudopneumoniae* were not detected as *S. pneumoniae* as it is specified in the producer’s data. QS and FA producer’s data both report cross reactivity between *Haemophilus haemolyticus* and *Haemophilus influenzae*. Given our results and the producers’ data, the specificity of these two POC SPCR seems equivalent.

Among all CSF samples tests performed with QS (*n* = 48) and FA (*n* = 37), 3 false positive results were obtained with QS and 1 with FA, corresponding to a specificity of 93.8% and 97.3%, respectively. The QS false positives results for *Hi* with a 37.5 Ct value and HSV-1 with a 35.6 Ct value and the FA false positive result for HSV-1 would probably be due to contaminations during handling. Indeed, suspensions of *Hi* and HSV-1 had been prepared near the POC SPCR assays preparation site shortly before. These results are a reminder of the vulnerability of POC SPCR assays to contaminations, due to their implementation in a not dedicated area to molecular biology [20,21]. A reserved cabinet was used for the following assays. The QS uses a one-step cartridge and FA a two-step. Thus, a one-step may help to reduce sample contamination and hence false positive results. The QS false positive result for HHV-6 with a 37.3 Ct value may be due to the presence a very small amount of HHV-6 DNA in the whole blood added for the interference test, below the positivity threshold of the R-GENE^®^ PCR assay, or a false positive due to the use of RUO cartridges. In fact, the production lines of RUO cartridges would be less controlled than those of CE-IVD ones. In daily practice, the availability of Ct values in the QS system, indicating a high Ct value can likely be alerted of possible false positive results. This is an important advantage over FA, which provides only qualitative results. Of note, the QS raw data is not available and replaced by curves improved by the manufacturer. Previous studies of POC PCR have shown the interest of having access to the raw curves [22]. The 2 viral targets (HSV-1 and PeV) and the 2 bacterial targets (*Nm* and *Sp*) were tested positive with 3 cartridges of the 2 SPCR POC assays. QS Ct values were very close (Ct value variation below 2) which is in favor of a high reproducibility of this POC SPCR assay. Surprisingly, internal control Ct values obtained with series of cartridges loaded with same content were sometimes significantly scattered (Ct value variation below 2). This can be inconvenient to evaluate a mild inhibition, which could mask a low positive viral or bacterial target. The potentially interfering substances tested (leukocytes, whole blood, proteins) did not significantly impact the HSV-1 Ct values compared to the reference Ct value. This suggests a robustness of the QS POC SPCR assay towards interfering substances found in CSF samples. FA results were in agreement with what was expected for reproducibility and absence of interferences, but a detailed analysis could not be performed on these qualitative results (no Ct values available).

Only QS allows for the detection of GAS Meningitis which are uncommon among cases of meningitis [3,23].

The main strength of this study is to provide a detailed analysis of the performance characteristics of QS for the detection of both viruses and bacteria, thus giving a global vision of QS performance from a point of view independent of that of the manufacturer. This work has several limitations such as the small number of clinical samples included due to the limited number of cartridges available for this study. The absence of *Cryptococcus neoformans*/*gattii* among the included strains is another limitation of this work which should be completed by the study of QS performances for the detection of these pathogenic yeasts. As it has been done before with FA, further prospective and retrospective studies are needed to accurately study the clinical performances of QS for the diagnosis of meningitis/encephalitis infections.

Nosocomial bacterial meningitis are common events in hospitals within a neurosurgical unit, but they are mostly due to bacterial species which are different from those causing community-acquired meningitis [24] and therefore neither QS nor FA are prepared for their diagnosis. Such a “nosocomial” panel which would include most of the corresponding species could be a really forthcoming tool for physicians in this context.

## 5. Conclusions

QIAstat-Dx^®^ Meningitis/Encephalitis panel gave the expected results in terms of assays reproducibility, carryover contamination and interferences as well as with the clinical CSF samples included in this study. Concerning the low concentration samples, the QIAstat-Dx^®^ Meningitis/Encephalitis panel correctly detected all the targets except for *S. agalactiae* (detected at 3000 CFU/mL instead of 1000 CFU/mL), Enterovirus-D68-B3 (not detected) but this may not be of clinical significance, and Echovirus-30 (not detected at the 1/10 dilution). It will be interesting to follow the data concerning these pathogens in future clinical studies with QIAstat-Dx^®^ system and QIAstat-Dx^®^ Meningitis/Encephalitis panel for the diagnosis of meningitis/encephalitis infections. Overall, our results suggest a good performance of QS which could constitute a suitable tool for laboratories provided that it is used with caution to reduce the rate of false positive results. Defining a diagnosis algorithm may also optimize its usefulness. This recently developed POC SPCR assay has the advantage of availability of Ct values and amplification curves from a one-step cartridge and also proposes original and interesting targets (*M. pneumoniae* and *S. pyogenes*) for the bacterial panel.

## Figures and Tables

**Figure 1 diagnostics-13-01110-f001:**
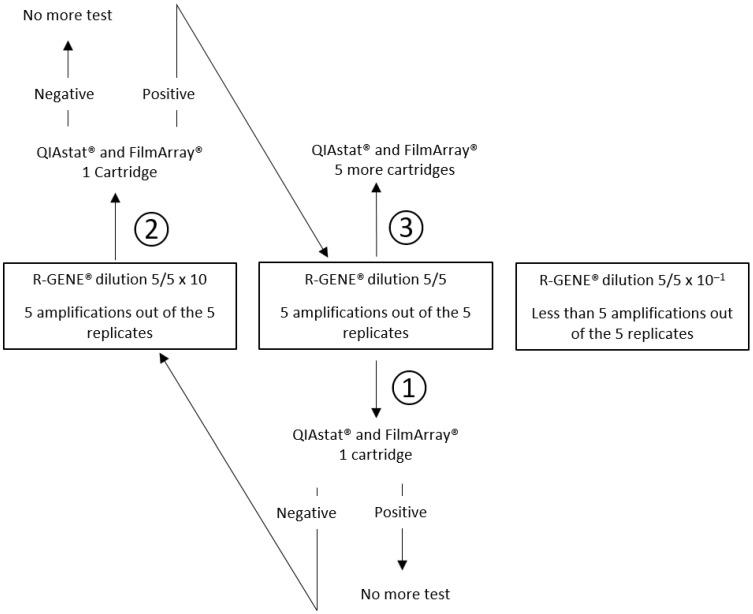
Strategy for assessing the detection of low viral and *Mp* loads, from step 1 to 3, using CSF diluted pool with positive results with R-GENE^®^ in 5 replicates and tested by two point-of-care syndromic PCR QIAstat Dx^®^ (QS) and FilmArray^®^ (FA).

**Table 1 diagnostics-13-01110-t001:** Bacterial strains used in the study.

Species	Origin Description
*Escherichia coli* K1	Clinical strain isolated from CSF
*Haemophilus influenzae* Type e	Clinical strain isolated from CSF
*Listeria monocytogenes* Type 4b	Clinical strain isolated from blood culture
*Neisseria meningitidis* Serotype B	Clinical strain isolated from CSF
*Mycoplasma pneumoniae*	M129
*Streptococcus agalactiae* ST17 clone	Clinical strain isolated from CSF
*Streptococcus mitis*	DSM 12643
*Streptococcus oralis*	DSM 20627
*Streptococcus pneumoniae*	ATCC 49619
*Streptococcus pseudopneumoniae*	DSM 18670
*Streptococcus pyogenes* Serotype M77	Clinical strain isolated from blood culture

**Table 2 diagnostics-13-01110-t002:** Detections of low viral and *Mp* loads with R-GENE^®^ PCR assays, QS and FA.

Target	Dilution	Detected by R-GENE^®^	R-GENE^®^ Median Ct Value	R-GENE^®^ Median Titer (copies/mL)	Detected by FA	Detected by QS	QS Ct Value
EV-A71-C1	1	5/5	34.99	ND	ND	1/1	36.0
	1/10	5/5	37.7	ND	1/1	3/6	38.5/38.7/38.5
	1/100	3/5	38.22	ND	ND	ND	ND
EV-D68-B3	1	5/5	32.71	ND	1/1	0/1	ND
	1/10	5/5	35.57	ND	5/6	0/1	ND
	1/100	4/5	40	ND	ND	ND	ND
E-30	1	5/5	34.52	ND	ND	1/1	37.5
	1/10	5/5	37.49	ND	1/1	0/6	ND
	1/100	3/5	40	ND	ND	ND	ND
E-6	1/10	5/5	38.58	ND	1/1	1/1	38.5
	1/100	3/5	40	ND	ND	ND	ND
PeV-1	1/10	5/5	38.41	ND	1/1	1/1	33.4
	1/100	3/5	40	ND	ND	ND	ND
PeV-3	1/10	5/5	37.52	ND	1/1	1/1	35.0
	1/100	3/5	40	ND	ND	ND	ND
HSV-1	1/10	5/5	36.41	1280	1/1	1/1	35.4
	1/100	2/5	39.03	<250	ND	ND	ND
HSV-2	1/10	5/5	35.39	277	1/1	1/1	36.5
	1/100	3/5	37.61	<100	ND	ND	ND
HHV-6	1/10	5/5	35.83	534	1/1	1/1	37.2
	1/100	1/5	39.28	<200	ND	ND	ND
VZV	1/10	5/5	37.25	<300	1/1	1/1	35.6
	1/100	3/5	40	<300	ND	ND	ND
*Mp*	1/10	5/5	36.21	ND	ND	1/1	34.8
	1/100	2/5	39.32	ND	ND	ND	ND

Ct: cycle threshold; EV: Enterovirus; E: Echovirus; PeV: Parechovirus; HSV: Herpes simplex virus; HHV-6: Human herpesvirus 6; VZV: Varicella-zoster virus; *Mp*: *Mycoplasma pneumoniae*. ND: Not determined.

**Table 3 diagnostics-13-01110-t003:** Results of the detection of low bacterial concentrations with QS and FA.

Bacterial Target	Concentration (CFU/mL)	Detected by QS	QS Ct Value	Detected by FA
*E. coli (Ec)*	1000	1/1	34.7	1/1
*L. monocytogenes (Lm)*	1000	1/1	36	1/1
*H. influenzae (Hi)*	1000	1/1	30	1/1
*N. meningitidis (Nm)*	100	1/1	34.1	1/1
*S. pneumoniae (Sp)*	100	1/1	35.6	1/1
	200	1/1	34.7	1/1
*S. agalactiae* (GBS)	1000	0/1	ND	1/1
	1750	0/1	ND	ND
	3000	1/1	35.9	ND
*S. pyogenes* (GAS)	1000	1/1	38.2	ND

ND: Not determined.

**Table 4 diagnostics-13-01110-t004:** QS and FA false positive results.

	Assays (*n*)	False Positive Assays (*n*)
QS	48	3
FA	37	1

**Table 5 diagnostics-13-01110-t005:** Clinical CSF samples results with QS and FA.

	Routinely-Used PCR Ct Value	Detected by QS	QS Ct Value	Detected by FA
CSF + *N. meningitidis*	ND	1/1	31.3	1/1
CSF *+ S. pneumoniae*	ND	1/1	19.8	1/1
CSF + HSV-2	37.2	1/1	36.5	1/1
CSF + VZV	33.7	1/1	33.0	1/1
CSF + PeV	34.3	1/1	31.9	1/1

ND: Not determined.

## Data Availability

Not applicable.

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
