# Peer review of "Evaluation of the Performance Characteristics of a New POC Multiplex PCR Assay for the Diagnosis of Viral and Bacterial Neuromeningeal Infections"

_diagnostics, 2023, doi:10.3390/diagnostics13061110_

Round 1

Reviewer 1 Report

Dear authors!

The work you have done and the well-structured manuscript are undeniably necessary and will help to make choices in clinical laboratories in order to discover the etiopathogenetic factors of the disease as efficiently and in a short time as possible.

However, very minor improvements/additions are needed:

1. 2.3. and 2.5.4. pay attention to text formatting

2. To make the design of the research easier to understand, I recommend creating a schematic diagram (point 2.5.1)

3. Table 2 - name of the virus - Varicella zoster (please correct what is written now) and put all the necessary punctuation marks and spaces correctly.

4. You refer to your data in several places in the discussion. Are they available in open databases? Please note.

5. Presentation of literature references in a uniform style is required.

6. Several literature references on herpes virus detection are very old - please supplement/replace with current references.

Author Response

Brest, 21st December

Dear reviewer,

We thank you for the pertinent comments you have made on our work and which have helped to improve our manuscript.

We have answered point by point and taken into account all your suggestions.

Léa Pilorgé

Reviewer 2 Report

Dear Authors 

Thank you for your interesting manuscript "Evaluation of the analytical performances of a new POC multiplex PCR assay for the diagnosis of viral and bacterial neuromeningeal infections" 

However some issues need to be addressed: 

- Introduction is very small it needs more details about the disease, the causative agent and the defect of the current methods and how it affect the patients

- Source of the isolates must be mentioned 

- Journal guidelines must be followed accurately 

- The ethical approval statement of the samples is lacking 

- Statistical analysis is needed

- Minor English revision is needed

- Actual results must be mentioned in the conclusion 

- Lines and styles must be revised carefully 

- More recent references and methods must be included 

Author Response

(The authors gave the same response as above.)

Reviewer 3 Report

Qualitative multiplexed nucleic acid-based diagnostic tests capable of simultaneous detection and identification of multiple bacterial, viral, and yeast nucleic acids from CSF specimens have become a fundamental element of the routine diagnostic laboratory. While the FilmArray ME panel has been on the market for several years, the QIAstat-Dx ME panel has been introduced recently. Performance studies are especially important to those readers considering integration of such a test into their laboratory’s diagnostic portfolio. The study by Le Bars et al. represents the first head-to-head performance evaluation of the panels mentioned above.

Comments:

(1)      The study consists of a large analytical study based on a spiked negative CSF pool and a very small clinical study based on 5 clinical CSF samples only. Why has only “analytical performance” been mentioned in the title, the abstract, and the objective (introduction, last paragraph)?

(2)      Abstract: What do authors mean with “unexpected” results? Were those false-positives or false-negatives?

(3)      Materials and Methods: The R-Gene assays were used as ref method; however, for detection of HSV in one of the clinical samples, another assay, the BioGX assay for use on the BD instrument which has never been published in a peer-reviewed international journal, was used. Authors should report the result obtained by the R-gene assay.

(4)      Materials and Methods: What does “inter-sample contamination” mean? This term does not exist in scientific language. “Biological interference”: “biological” should be skipped as non-biological substances may also interfere with the test system. “Native CSF samples” should be replaced by “Clinical samples”.

(5)      Discussion: The discussion section should be shortened significantly as it mainly repeats data already shown in the results subsections without an adequate interpretation.

(6)      Discussion: A paragraph discussing the strengths and the limitations of the study must be included.

(7)      English must be improved significantly. There are numerous linguistic bugs (especially with grammar) and several typos.

Author Response

Dear Reviewer,

We thank you very much for your attentive reading and the relevance of your remarks.

We have taken all of your remarks into account to improve the quality of our manuscript and we have answered them point by point (in blue).

With kind regards,

Léa Pilorgé

Reviewer 4 Report

In this study, the authors want to verify the QIAstat-Dx (on going proven test kit) with the available commercial kit (BioFire Film array:FA) to detect bacteria and virus causing meningitis in CSF.

               This is an interesting research to provide the information about an alternative test kit to detect bacterial and viral meningitis. Anyway, there are some points to clarified/minor editing.

1.      For the cross-reactivity test, other closely related of viral and also bacteria (more than 3 species) should be tested. 

2.      In the result, table 2 could it be presented as viral copies/mL instead of the dilution.

3.      In the result, topic 3.3 the unit should be CFU/mL (In CSF pool spiked with 200 UFC/mL…)

4.      Are there any report about the case of viral and bacterial mix infection meningitis? Have you ever tested the mix infection (spike) with the QS?

5.      For the native samples tested, 5 sample seem to be low to validates the QS. Are there any more samples? If not, you should suggest that more native samples should be evaluated in the discussion too.

6.      In abstract and all through the article, the bacterial genus and species should be in italic.

Author Response

(The authors gave the same response as above.)
